# LoRaQ: Optimized Low Rank Approximation for 4-bit Quantization

## Abstract

Post-training quantization (PTQ) is essential for deploying large diffusion-based transformers on resource-constrained hardware. However, aggressive 4-bit quantization introduces significant degradation in generative performance. While existing solutions mitigate quantization error through outlier smoothing or rotation techniques, low-rank approximation methods that add auxiliary linear branches to each quantized layer represent a promising new paradigm. Yet, these approaches suffer from computational overhead due to the data movement required by full-precision (W16A16) branches, limiting practical deployment. In addition, data-based calibration contributes to the computational complexity of the quantization process, especially because search policies must evaluate many parameter configurations using a small calibration subset. We propose LoRaQ (low-rank approximated quantization), a data-free calibration approach to optimize quantization error compensation. This method can be used in composition with other PTQ models. LoRaQ further enables mixed-precision configurations by quantizing the low-rank branch itself, overcoming the limitations of prior work. While LoRaQ achieves superior quantization performance than state-of-the-art methods in their native W4A4 setting on PixArt-$\Sigma$ and SANA, it also allows for configurations such as W8A8,W6A6 and W4A8 for low-rank branch alongside a W4 main layer. This reduces data movement overhead and enables a fully quantized, hardware-efficient solution.

## 1 Introduction

The proliferation of large-scale generative models, particularly diffusion-based transformers such as Black-Forest-Labs (2024); Esser et al. (2024), has precipitated an urgent need for efficient inference strategies. Quantization has emerged as a pivotal technique for reducing computational and memory costs, thereby enabling the deployment of these models on diverse hardware platforms, including edge devices, consumer platforms, and data center accelerators. However, aggressive quantization often leads to pronounced degradation in generative fidelity, primarily due to the sensitivity of these architectures to minute perturbations in weights and activations and their iterative denoising.

Recent advances have sought to ameliorate quantization-induced errors through channel-wise scaling and low-rank approximations (Xiao et al., 2023; Li et al., 2025). In particular, low-rank branches appended to quantized linear layers have demonstrated the capacity to preserve critical information at higher precision, justifying the use of 16-bit representations for these components. However, the retention of full-precision activations in these branches requires 16-bit operations such as matrix multiply and data movement of 16-bit activations. This introduces substantial computational overhead, exacerbated by increased data movement and limited by the scope of fused GPU kernels. This bottleneck impedes the generalization of such methods to broader hardware ecosystems and model architectures.

Furthermore, prevailing approaches rely on resource-intensive, data-dependent calibration to find quantized representations. The lack of lightweight alternatives imposes a substantial computational overhead, creating a barrier for quantizing large-scale models with limited resources and complicating integration into production workflows. In this work, we address these limitations by introducing a data-free optimization framework that directly minimizes quantization error via the low-rank branch, enabling more aggressive and efficient quantization of both weights and activations.

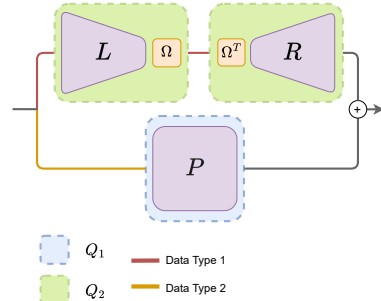

Figure 1: Overview of the LoRaQ pipeline. A linear layer's weight $\boldsymbol{W}$ is decomposed into two parallel branches. The residual branch contains the quantized residual matrix $\boldsymbol{P} = \boldsymbol{W} - Q_2(\boldsymbol{L\Omega})Q_2(\boldsymbol{\Omega}^T\boldsymbol{R})$, quantized with operator $Q_1$. The low-rank branch contains the matrices $\boldsymbol{L}$ and $\boldsymbol{R}$, which are rotated by $\boldsymbol{\Omega}$ and quantized using operator $Q_2$. We show that inserting $\boldsymbol{\Omega}$ between $\boldsymbol{L}$ and $\boldsymbol{R}$ minimizes quantization error. The rotation matrix is fused with the low-rank matrices and has no extra overhead at inference.

Our contributions are threefold. First, we present the first post-training quantization (PTQ) scheme for transformer-based diffusion models (DiTs) that achieves sub-8-bit quantized path for both weights and activations without floating scales, maintaining high generative quality. Second, we propose a mixed-precision quantization strategy, leveraging micro-scaling formats and block-wise power-of-two scales to enhance computational efficiency, particularly for GEMM-based architectures. Third, we release an open-source, hardware-agnostic PTQ library for transformer blocks, facilitating systematic benchmarking across methods and configurations and supporting scalable quantization of large models in multi-GPU environments.

After reviewing related work, we introduce our method, LoRaQ represented in Figure 1, which splits a linear layer into a low-rank and a residual branch. Our approach optimizes the low-rank matrices, which in turn defines the residual weights and enables the quantization of the low-rank matrix itself. In the experiments section, we compare LoRaQ against SVDQuant using equivalent configurations for a fair comparison across different datasets, models, and metrics. We also analyze various mixed-precision configurations of our method, considering the support for such operations in modern hardware.

## 2 RELATED WORK

Diffusion models have rapidly advanced and transformed text-to-image synthesis (Sohl-Dickstein et al., 2015; Ho et al., 2020). They generate high-quality samples via an iterative denoising process starting from Gaussian noise. Recently, the core architecture shifted from U-Net (Ronneberger et al., 2015) to transformer-based backbones known as DiTs, as pioneered by Peebles & Xie (2023) and Bao et al. (2023). Enabled by the scalability of the transformer backbone (Vaswani et al., 2017), this shift opened new opportunities for higher image quality. Key advances include MM-DiT (Esser et al., 2024) and the FLUX.1 suite of models (Black-Forest-Labs, 2024) for efficient scaling and high-resolution image synthesis; and PixArt-$\alpha$ (Chen et al., 2024b) and PixArt-$\Sigma$ (Chen et al., 2024a), which target training strategies and architecture efficiency.

The significant potential of DiTs has motivated the deployment of these models on many resource-constrained devices. However, DiTs require substantial computation to achieve high image quality (Xie et al., 2025; Li et al., 2025), which makes deployment challenging, particularly with efficient narrow data types. We focus on overcoming this barrier to enable low-precision inference on resource-constrained devices while preserving image generation quality.

Quantizing DiTs presents several new challenges. Modern GPUs' computational constraints necessitate both weight and activation quantization, which is complicated by substantial data variation across token, condition, timestep, and channel dimensions. Early quantization efforts struggled with systematic activation outliers which contributed to a degradation of accuracy (Dettmers et al., 2022; Wei et al., 2023). A notable breakthrough was achieved by Xiao et al. (2023), enabling efficient 8-bit weight quantization (W8) and 8-bit activation quantization (A8) through the transfer of quantization difficulty from activations to weights using per-channel scaling transformations. Employing these techniques for diffusion models has evolved from 8-bit methods like PTQ4DM (Shang et al., 2023) to more advanced strategies to employ sub-8-bit data types for the weights such as timestep-aware calibration (Li et al., 2023), sample-wise dynamic activation quantization (Chen et al., 2024c), salient channels (Wu et al., 2024), sensitivity-aware quantization (Yang et al., 2023), timestep-conditioned methods (He et al., 2023); (Huang et al., 2024), as well as extensions to both image (Tang et al., 2024; Zhao et al., 2024) and video models (Zhao et al., 2025).

Aggressive 4-bit quantization (W4A4) requires new strategies to overcome outlier sensitivity. Ashkboos et al. (2025); Liu et al. (2025); Tseng et al. (2024) introduce the use of rotations matrices to handle outliers of the activations. Zhang et al. (2025) achieved micro-scaling FP4 attention for inference. Li et al. (2025) consolidated outliers from activations to weights via smoothing, then decomposed updated weights using SVD into high-precision low-rank and 4-bit quantized residual branches, with fused kernels eliminating memory access overhead for practical W4A4 deployment on GPUs.

Our method improves offline quantization through calibration-free optimization that can be seamlessly integrated into existing quantization approach such as Xiao et al. (2023) and Liu et al. (2025). Unlike Li et al. (2025)'s approach of maintaining a full-precision low-rank branch, we propose a quantized low-rank branch to reduce data movement overhead, eliminating custom kernel requirements and enabling flexible mixed-precision weight matrices. In addition, we show that the combination of higher ranks with narrower data formats improves the model's performance.

## 3 METHOD

### 3.1 QUANTIZATION ERROR AND UPPER BOUND

Following the formulation of Li et al. (2025), we examine a linear transformation with input matrix $\boldsymbol{X} \in \mathbb{R}^{m \times d}$ and weight matrix $\boldsymbol{W} \in \mathbb{R}^{d \times n}$, where $m$, $n$, and $d$ correspond to the dimensions of input, output and hidden features, respectively. We can express the quantization error as

$$\mathcal{E}(\boldsymbol{X}, \boldsymbol{W}) = \|\boldsymbol{X}\boldsymbol{W} - Q(\boldsymbol{X})Q(\boldsymbol{W})\|_F \, , \tag{1}$$

where $\|\cdot\|_F$ represents the Frobenius norm and $Q$ is a quantization operator. The quantization error admits the following upper bound:

$$\mathcal{E}(\boldsymbol{X}, \boldsymbol{W}) \leq \|\boldsymbol{X} - Q(\boldsymbol{X})\|_F \|\boldsymbol{W}\|_F + \|Q(\boldsymbol{X})\|_F \|\boldsymbol{W} - Q(\boldsymbol{W})\|_F. \tag{2}$$

See Appendix A.3.1 for the proof. This reveals that the quantization error is constrained by the quantization discrepancies in the original matrices $\|\boldsymbol{W} - Q(\boldsymbol{W})\|_F$ and $\|\boldsymbol{X} - Q(\boldsymbol{X})\|_F$. Our method focuses on reducing $\|\boldsymbol{W} - Q(\boldsymbol{W})\|_F$ in a more aggressive way than state-of-the-art methods (Xiao et al., 2023; Li et al., 2025) and is therefore meant to be implemented along existing smoothing methods that focus on reducing $\|\boldsymbol{X} - Q(\boldsymbol{X})\|_F$.

### 3.2 ABSORBING THE QUANTIZATION ERROR

Following the emerging paradigm of incorporating full-precision low-rank branches into quantized linear layers to mitigate quantization error (Li et al., 2025), we propose a novel approach that directly approximates the quantization error using a low-rank matrix that can be quantized. We argue that this strategy is more effective than approximating the weight matrix itself with a low-rank decomposition, as our method provides explicit quantization error compensation, which adapts to the quantization operator and the chosen data format. Intuitively, the quantization function introduces structure in the error space that previous methods fail to exploit.

Specifically, we seek to find a new point in the input space of the quantization function that shapes the quantization error to a low-rank matrix such that the quantization error can be expressed as

$$\hat{\boldsymbol{W}} - \boldsymbol{W} = Q(\boldsymbol{W} + \boldsymbol{D}) - \boldsymbol{W} = \boldsymbol{L}\boldsymbol{R} \tag{3}$$

where $\hat{\boldsymbol{W}}$ is the quantized version of the new point in the input space. $\boldsymbol{D} \in \mathbb{R}^{d \times n}$ is a perturbation matrix, $\boldsymbol{L} \in \mathbb{R}^{d \times \rho}$ and $\boldsymbol{R} \in \mathbb{R}^{\rho \times n}$ for a fixed $\rho << d, n$. The key insight is that by strategically shifting the original weight matrix $\boldsymbol{W}$ with $\boldsymbol{D}$, we can obtain a quantization error that exhibits inherent low-rank structure, thereby enabling efficient error approximation.

We can express the search of the new point as a gradient descent optimization problem where we minimize the difference between the quantized version of the perturbed weight matrix $\hat{\boldsymbol{W}}$ and the original weight matrix $\boldsymbol{W}$, subject to the constraint that the error is low-rank. However, considering that the derivative of the quantization function is zero almost everywhere, we cannot directly compute the gradient of the quantization function with respect to the perturbation matrix $\boldsymbol{D}$.

Instead, we can reformulate the problem by considering the quantization function as an idempotent operator. By definition, it means that applying the quantization function twice is equivalent to applying it once: we define the image of $Q$ as $\mathbb{I}$, then $Q(\boldsymbol{X}) = \boldsymbol{X}, \forall \boldsymbol{X} \in \mathbb{I}$. According to Equation 3, the equality implies that $(\boldsymbol{W} + \boldsymbol{LR}) \in \mathbb{I}$. Thus, $Q(\boldsymbol{W} + \boldsymbol{LR}) = \boldsymbol{W} + \boldsymbol{LR}$. Using this property, it is sufficient to find a low-rank perturbation matrix $\boldsymbol{D} = \boldsymbol{LR}$ in order to approximate the quantization error as a low-rank matrix.

Thus, we propose to solve the optimization problem

$$
\begin{aligned}
\boldsymbol{L}^*, \boldsymbol{R}^* &= \arg\min_{\boldsymbol{L},\boldsymbol{R}} \|Q(\boldsymbol{W} + \boldsymbol{LR}) - \boldsymbol{W} - \boldsymbol{LR}\|_F \\
&\propto \arg\min_{\boldsymbol{L},\boldsymbol{R}} \mathcal{L}(Q(\boldsymbol{W} + \boldsymbol{LR}) - \boldsymbol{W}, \boldsymbol{LR}) ,
\end{aligned}
\tag{4}
$$

where $\mathcal{L}$ is the Mean Squared Error loss.

Indeed, using this approximation of $\boldsymbol{D}$, we can now compute a gradient of our loss function with respect to $\boldsymbol{L}$ and $\boldsymbol{R}$ as detailed in Appendix A.3.2.

This allows us to iteratively update the low-rank matrices $\boldsymbol{L}$ and $\boldsymbol{R}$ using gradient descent, effectively absorbing the quantization error into a low-rank structure that can be efficiently handled during inference with a fixed rank defined. This rank can be chosen based on the desired trade-off between computational efficiency and quantization error correction, allowing for flexible deployment and further improvements.

We initialize $\boldsymbol{L}$ and $\boldsymbol{R}$ using the Singular Value Decomposition (SVD) of $\boldsymbol{W}$. This allows us to find the matrix of a predefined rank $\rho$, $\boldsymbol{M} = \boldsymbol{L}_0 \boldsymbol{R}_0$, $\boldsymbol{L}_0 \in \mathbb{R}^{d \times \rho}$, $\boldsymbol{R}_0 \in \mathbb{R}^{\rho \times n}$ such that $\boldsymbol{M} = \min_{\boldsymbol{B}/\mathrm{rank}(\boldsymbol{B})=\rho} \|\boldsymbol{W} - \boldsymbol{B}\|_F$. As Li et al. (2025), we found it to be a good initialization for our optimization algorithm, where $\boldsymbol{L} = -\boldsymbol{L}_0$ and $\boldsymbol{R} = \boldsymbol{R}_0$.

### 3.3 QUANTIZING HIGHER RANK MATRICES

Considering the data formats available in hardware architectures (Microsoft et al., 2023; Li et al., 2025; Zhang et al., 2025), we can define a budget $\beta$ as the maximum allowable memory and computation resources for the low-rank branch in our approximation. $\beta$ is defined by the number of bits we require per channel for the low-rank branch. For example, SVDQuant uses a float16 low-rank branch with $\rho = 32$ for its 4-bit quantization, thus allocating a budget of $\beta = 16 \text{ bits/value} \times 32 \text{ values/channel} = 512 \text{ bits/channel}$ for the low-rank branch.

In our case, optimizing a low-rank matrix to find a solution easily leads to a local minimum. However, the higher the rank of the matrix, the better the lower local minimum we find. To respect a budget $\beta$ and leverage the benefits of higher rank representations, we also suggest quantizing the low-rank branch. We find that the accuracy gains from using a higher rank outweigh the loss from quantizing the low-rank matrices.

Reminding the description of the method in Figure 1, we now consider two quantization functions $Q_1$ for the residual branch and $Q_2$ for the low-rank branch to be represented in $n < 16$ bits/value such that

$$
Q_1(\boldsymbol{W} + Q_2(\boldsymbol{L})Q_2(\boldsymbol{R})) - \boldsymbol{W} \approx Q_2(\boldsymbol{L})Q_2(\boldsymbol{R}) ,
\tag{5}
$$

where $\boldsymbol{L} \in \mathbb{R}^{d \times \rho'}$ and $\boldsymbol{R} \in \mathbb{R}^{\rho' \times n}$ with $\rho' = \mathrm{floor}(\frac{\beta}{n})$.

While this quantization leads to the intrinsic degradation of the low-rank branch, we will show that it does provide significant improvements in approximating $\boldsymbol{W}$ in the quantized space thanks to the increased rank of the branch.

**Rotation aware method** To maximize the performance of the method, inspired by Liu et al. (2025) and their Cayley SDG method, we mitigate the quantization error induced by $Q_2$ without any memory or computation overhead by using a rotation-aware optimization method. We define and optimize a rotation matrix $\boldsymbol{\Omega} \in \mathbb{R}^{\rho' \times \rho'}$ for each low-rank branch and attempt to solve the following equation, knowing the optimal low-rank matrices $\boldsymbol{L}^*$ and $\boldsymbol{R}^*$:

$$
\boldsymbol{\Omega}^* = \min_{\boldsymbol{\Omega}}(\|Q_2(\boldsymbol{L}^*\boldsymbol{\Omega}) - \boldsymbol{L}^*\boldsymbol{\Omega}\|_F + \|Q_2(\boldsymbol{\Omega}^T\boldsymbol{R}^*) - \boldsymbol{\Omega}^T\boldsymbol{R}^*\|_F) .
\tag{6}
$$

Thus, we define the following loss function:

$$\mathcal{L}_{\boldsymbol{\Omega}} = \mathcal{L}(Q_2(\boldsymbol{L}^*\boldsymbol{\Omega}), \boldsymbol{L}^*\boldsymbol{\Omega}) + \mathcal{L}(Q_2(\boldsymbol{\Omega}^T\boldsymbol{R}^*), \boldsymbol{\Omega}^T\boldsymbol{R}^*) \,. \tag{7}$$

### 3.4 QUANTIZING ACTIVATIONS

State-of-the-art methods consider an aggressive 4-bit quantization of the inputs of the linear layer in each transformer block of the quantized model. As a consequence, a direct quantization of the activations is not possible without a significant drop in accuracy. Following Li et al. (2025); Liu et al. (2025), we adopt activation smoothing as a technique complementary to ours for activation quantization.

## 4 EXPERIMENTS

### 4.1 SETUPS

**Models** We quantize SANA (Xie et al., 2025) with 1.6 billion parameters and PixArt-$\Sigma$ (Chen et al., 2024b) with 0.6 billion parameters.

**Datasets** Following previous work (Li et al., 2025), we benchmark our quantization methods on the MJHQ-30K and sDCI (Li et al., 2024; Urbanek et al., 2024) datasets. We draw 5000 samples for each dataset.

**Data Formats** We benchmark against baselines using SINT4 and also evaluate on OCP Microscaling (MX) Formats (Microsoft et al., 2023), detailed in Appendix A.2.1. Our method flexibly combines rank and bit-width for the low-rank branch, using various MX types (MXFP8e4, MXFP6e2, MXFP4e2, MXINT8, MXINT4) without mixing INT and FP formats within a layer. MX formats leverage efficient power-of-two scaling and are supported by modern hardware like the AMD MI350/355 (Advanced Micro Devices, 2025), enabling native mixed-precision operations for which our method is designed. We use the TensorCast library (Dellinger & Khodamoradi, 2025) for all quantization procedures.

**Baselines** We compare our method against two baselines. For SANA, we use a simple round-to-nearest (RTN) quantization strategy. Our primary baseline is the state-of-the-art method SVDQuant (Li et al., 2025), which consistently outperforms other methods like Zhao et al. (2025) on PixArt-$\Sigma$ across various models and datasets.

**Metrics** Following existing benchmarks, we evaluate performance on two criteria. To measure similarity to the 16-bit baseline, we use Learned Perceptual Image Patch Similarity (LPIPS) (Zhang et al., 2018) and Peak Signal-to-Noise Ratio (PSNR). To assess overall visual quality, we use Frechet Inception Distance (FID) (Heusel et al., 2018), Image Reward (IR) (Xu et al., 2023), and Kernel Inception Distance (KID) (Bińkowski et al., 2021).

**Experimental Organization** We first compare our calibration method for weight quantization to SVDQuant's in a W4A4 setting using the same data formats. As these results consider the activations for the low-rank branches at full precision, we then analyze various sub-16-bit mixed-precision configurations with our method, leveraging its flexibility to quantize low-rank matrices. We begin by analyzing the trade-off between rank and bit-width within a fixed memory budget for the low-rank matrices, while keeping activations at 8-bit precision. Subsequently, we configure both the activations and the low-rank branch to be quantized to the same sub-16-bit data format and analyze performance with respect to the rank, as this allows for larger low-rank matrices without increasing latency. Finally, we analyze the influence of the rotation matrix we insert in the low-rank branch for quantizing the low-rank matrices with minimal error.

### 4.2 MAIN RESULTS

#### 4.2.1 QUANTITATIVE RESULTS

We report the results in Table 1 and Table 2, which show that at equivalent data format and dataset, our method outperforms the baseline methods. We detail the configurations of SVDQuant and Lo-RaQ in Appendix A.1.1. LoRaQ relies on the same smoothing calibration (Xiao et al., 2023) as

Table 1: Quantitative quality comparisons across different models. Following SVDQuant, we use SINT4 to ensure a fair comparison with methods that consider the data format as part of the method.

| Model | Format | Precision (W-A) | Method | MJHQ | | | | sDCI | | | |
| | | | | Quality | | Similarity | | Quality | | Similarity | |
| | | | | FID (↓) | IR (↑) | LPIPS (↓) | PSNR (↑) | FID (↓) | IR (↑) | LPIPS (↓) | PSNR (↑) |
| PixArt-Σ (20 Steps) | FP16 | 16-16 | – | 16.6 | 0.944 | – | – | 24.8 | 0.966 | – | – |
| | SINT4 | 4-4 | VIDIT-Q | 412 | -2.27 | 0.854 | 6.44 | 425 | -2.28 | 0.838 | 6.70 |
| | | | SVDQuant | 19.2 | 0.878 | 0.323 | **17.6** | 25.9 | 0.918 | 0.352 | **16.5** |
| | | | Ours | **16.9** | **0.898** | **0.309** | 17.6 | **24.1** | **0.919** | **0.346** | 16.2 |
| SANA-1.6B (20 Steps) | BF16 | 16-16 | – | 16.2 | 1.10 | – | – | 22.4 | 1.07 | – | – |
| | SINT4 | 4-4 | RTN | 20.5 | 0.894 | 0.339 | 15.3 | 28.6 | 0.807 | 0.371 | 13.8 |
| | | | SVDQuant | 19.3 | 0.935 | 0.220 | 17.8 | 28.1 | 0.846 | 0.242 | 16.2 |
| | | | Ours | **16.1** | **1.09** | **0.182** | **19.2** | **21.8** | **1.06** | **0.208** | **17.4** |

Table 2: Quantitative quality comparisons on PixArt-Σ. Using the MXINT4 format as the 4-bit precision, we show that LoRaQ outperforms our 4-bit baseline with power-of-two scale, showing that our method is data format agnostic and that our library can easily experiment on other data formats.

| Model | Format | Precision (W-A) | Method | MJHQ | | | | sDCI | | | |
| | | | | Quality | | Similarity | | Quality | | Similarity | |
| | | | | FID (↓) | IR (↑) | LPIPS (↓) | PSNR (↑) | FID (↓) | IR (↑) | LPIPS (↓) | PSNR (↑) |
| PixArt-Σ (20 Steps) | FP16 | 16-16 | – | 16.6 | 0.944 | – | – | 24.8 | 0.966 | – | – |
| | MXINT4 | 4-4 | SVDQuant | 18.9 | 0.738 | 0.424 | 15.9 | 26.1 | 0.902 | 0.435 | 14.7 |
| | | | Ours | **15.4** | **0.901** | **0.339** | **16.8** | **23.7** | **0.943** | **0.374** | **15.6** |

SVDQuant, but improves the quantization of the weights, which explains this consistent improvement following our observations from Equation 2. Additionally, SVDQuant uses an unsigned data type for the quantization of the inputs of a linear layer if a non-linear activation like ReLU precedes it. If any input of that layer has a negative lower-bound, the inputs are shifted to a positive range before quantization to use the unsigned data type. For fair comparison, in Table 1, we reproduced the quantization scheme in Li et al. (2025), but ignored this method in any other experiment. Thus, the increase in performance by LoRaQ is more important in Table 2 as no activation shifting is used for MX formats.

When interpreting these results, we distinguish between image quality and image similarity. For image quality metrics like FID, which compares the distribution of generated images against the ground-truth dataset, our method is generally much closer to the full-precision models. This suggests that our quantization method's performance is not strictly bounded by the image quality of the full-precision model. We also observe better performance with the Image Reward (IR) metric. Regarding image similarity (PSNR, LPIPS), our results are closer to SVDQuant, which can be explained by the constraints we imposed for a fair comparison: using a single sub-16-bit data format and maintaining an equivalent memory footprint for the low-rank overhead. An exception is noted for PixArt-Σ on the sDCI dataset, where our PSNR is slightly lower than SVDQuant's. We attribute this to minor local differences arising from our dataset-free weight quantization, which, however, does not negatively impact overall image quality and, as suggested by the FID score, may even improve it.

Finally, the significant performance improvement of LoRaQ on SANA can be explained by architectural differences. SANA features larger weight matrices, for which SVDQuant's fixed rank of 32 is too limiting to capture sufficient information for a small residual error. In contrast, LoRaQ's higher effective rank of 128, achieved at an equivalent memory footprint, allows it to capture more dimensions, leading to a more accurate representation.

### 4.2.2 VISUAL QUALITY

We provide some examples for the configurations of SVDQuant and LoRaQ used in Section 4.2.1. We respectively show in Figure 2 and Figure 3 some visual representations of the results from Table 1 and Table 2. Consistently, we observe that our method helps reduce the visual difference between the image generated by the full-precision model and the quantized model. Our method is

Figure 2: Comparison of images generated by PixArt-Σ in different configurations: Full precision model (FP16), SVDQuant with SINT4 residual branch quantization, and LoRaQ with SINT4 residual branch and low-rank matrices quantization.

| **FP16** | **SVDQuant** | **Ours** | **FP16** | **SVDQuant** | **Ours** |
|---|---|---|---|---|---|

Figure 3: Comparison of images generated by PixArt-Σ in different configurations: Full precision model (FP16), SVDQuant with MXINT4 residual branch quantization, and LoRaQ with MXINT4 residual branch and low-rank matrices quantization.

| **FP16** | **SVDQuant** | **Ours** | **FP16** | **SVDQuant** | **Ours** |
|---|---|---|---|---|---|

able to generate images that are more detailed and closer to the original than SVDQuant, which is consistent with the quantitative results we report in Section 4.2.1. Additional visual results are provided in Appendix A.4.

## 4.3 MIXED PRECISION ANALYSIS

Motivated by the growing hardware support for mixed-precision kernels (Advanced Micro Devices, 2025), we analyze various MX format configurations to evaluate their impact on model performance.

### 4.3.1 RANK VS. BIT-WIDTH TRADE-OFF AT A FIXED MEMORY BUDGET

We fix the activations of the low-rank branch to 8-bit precision to analyze the impact of rank and bit-width trade-offs under a constant activation quantization setting. The detailed configuration for each experiment is provided in Appendix A.2. This ensures a consistent comparison across all setups and highlights the robustness of LoRaQ under practical mixed-precision constraints.

We also report additional LoRaQ configurations to demonstrate robustness across a wider range of settings. For each setup, we adjust the low-rank branch's rank and bit width so that the effective bit budget (bits per value × rank) matches the SVDQuant baseline, guaranteeing relevance to applications. The bit budget succinctly quantifies the memory overhead introduced by the low-rank branch.

Table 3 demonstrates that, beyond the absolute results, how different rank/bit width trade-offs impact model performance, measured using various metrics. Using integer data formats, the larger number

Table 3: Mixed Precision Analysis : Evaluating the influence of the trade-off rank/precision with a fixed memory budget for the low-rank branch. The activation is at a 8-bit precision on both low-rank and residual branches.

| Model | Format (Residual) | Format (Low Rank Branch) | Rank | MJHQ | | | | | sDCI | | | | |
|---|---|---|---|---|---|---|---|---|---|---|---|---|---|
| | | | | Quality | | | Similarity | | Quality | | | Similarity | |
| | | | | FID ($\downarrow$) | KID ($10^{-3}$) ($\downarrow$) | IR ($\uparrow$) | LPIPS ($\downarrow$) | PSNR ($\uparrow$) | FID ($\downarrow$) | KID ($10^{-3}$) ($\downarrow$) | IR ($\uparrow$) | LPIPS ($\downarrow$) | PSNR ($\uparrow$) |
| PixArt-$\Sigma$ (20 Steps) | MXFP4e2 | FP16 | 32 | 16.8 | 1.86 | 0.934 | 0.349 | 16.4 | **22.1** | **4.06** | 0.962 | 0.390 | 14.9 |
| | | MXFP8e4 | 64 | 16.3 | 1.53 | 0.927 | 0.316 | 17.1 | 22.5 | 4.34 | 0.980 | **0.349** | **15.8** |
| | | MXFP6e2 | 86 | 16.4 | 1.62 | 0.961 | **0.313** | 17.2 | 22.4 | 4.20 | 0.985 | **0.349** | **15.8** |
| | | MXFP4e2 | 128 | **16.1** | **1.52** | **0.963** | 0.322 | 17.0 | 22.7 | 4.36 | 0.994 | 0.362 | 15.5 |
| | MXINT4 | MXINT8 | 64 | 16.4 | 1.66 | **0.935** | **0.275** | **18.0** | 23.5 | 4.88 | 0.952 | **0.305** | **16.7** |
| | | MXINT4 | 128 | **16.3** | **1.60** | 0.920 | 0.292 | 17.7 | 23.9 | 5.03 | 0.948 | 0.325 | 16.3 |

of mantissa bits in MXINT8 with a rank of 64 is the better choice for both MJHQ and sDCI. For mini-float numbers, higher ranks (86 and 128) work better for MJHQ, while sDCI metrics do not indicate a clear winner combination; instead, they suggest that the highest rank (128) can compete in image quality metric, IR, and the 6- and 8-bit mini-floats utilize their ranks better when measured for similarity metrics.

While some variability in performance exists across specific rank/bit width pairs, the results show that, starting from the SVDQuant configuration (rank=32 at 16 bits per value), LoRaQ can achieve an equivalent memory footprint with alternative rank/bit width pairs. In other words, LoRaQ maintains better performance across configurations in a sub-16-bit low-rank branch quantization, allowing practitioners to select a rank/bit width pairing that matches a target accelerator or latency constraint without sacrificing model quality, especially the performance of the low-rank branch of LoRaQ.

We provide visual examples for this experiment in Appendix A.4.

Table 4: Evaluating the effect of rank in LoRaQ with a fixed MXFP4e2 residual branch. We evaluate PixArt-$\Sigma$ on 5k samples from MKHQ dataset.

| Low Rank Branch and Activations | Rank | Quality | | | Similarity | |
|---|---|---|---|---|---|---|
| | | FID ($\downarrow$) | KID ($10^{-3}$) ($\downarrow$) | IR ($\uparrow$) | LPIPS ($\downarrow$) | PSNR ($\uparrow$) |
| MXFP8e4 | 32 | 16.6 | 1.64 | 0.912 | 0.365 | 16.6 |
| | 64 | 16.3 | 1.51 | 0.948 | 0.320 | 17.1 |
| | 96 | **16.1** | **1.41** | **0.966** | 0.311 | 17.2 |
| | 128 | 16.2 | 1.47 | 0.956 | **0.292** | **17.7** |
| MXFP6e2 | 32 | 16.8 | 1.82 | 0.889 | 0.374 | 16.5 |
| | 64 | 16.2 | 1.57 | 0.924 | 0.315 | 17.4 |
| | 96 | 16.3 | 1.56 | 0.926 | 0.321 | 17.5 |
| | 128 | **15.9** | **1.46** | 0.956 | **0.286** | **18.0** |

### 4.3.2 PERFORMANCE SCALING WITH RANK AT A FIXED PRECISION

With detailed configurations provided in Appendix A.2, this experiment emphasizes a key feature of our method: quantizing activations in the same data format as the low-rank branch. This unified sub-16-bit precision for both activations and weights allows us to explore higher ranks without the latency penalty of 16-bit operations. We analyze how increasing the rank impacts performance when both the low-rank weights and activations share the same quantization format.

Analyzing the results in Table 4, we observe that increasing the rank does not always lead to a linear increase in image quality metrics, although image similarity metrics do tend to improve more consistently with rank. This again highlights the distinction between perceptual quality and numerical similarity. More importantly, it demonstrates the robustness of our method across a range of ranks, showcasing that high performance can be achieved without maximizing rank. This flexibility opens up significant opportunities for hardware co-design, allowing practitioners to tune the rank for optimal latency on accelerators that support full sub-16-bit data format operations.

We provide visual examples for this experiment in Appendix A.4.

### 4.4 ABLATION STUDY

The key components of our LoRaQ pipeline, as shown in Figure 1, are calibrating the low-rank branch and inserting an optimized rotation matrix between the low-rank factors to minimize quanti-

zation error. To validate the impact of these components, we conduct an ablation study. In Table 5, we compare our full method against these ablated versions. Our low-rank branch is initialized via SVD decomposition and then optimized. We study the effect of optimized low-rank (as indicated in the Optimized LR column). We also study the impact of learned rotations by removing them (as indicated in the Rotations column). The results demonstrate that removing the rotation, as defined in Equation 6, consistently increases the quantization error. Our results also confirm that an optimized low-rank branch reduces the quantization error. This shows that the calibrated low-rank and optimized rotation are critical elements for achieving high precision in the quantized low-rank branch without incurring additional computational cost or memory overhead.

Table 5: Evaluating the influence of the optimized rotation regularization between the low rank matrices in LoRaQ. We evaluate PixArt-$\Sigma$ on 3k samples from MKHQ dataset

| Format (Low Rank Branch) | Rank | Optimized LR | Rotations | KID ($10^{-3}$) ($\downarrow$) | LPIPS ($\downarrow$) | PSNR ($\uparrow$) |
|---|---|---|---|---|---|---|
| MXFP8e4 | 64 | ✓ | ✓ | **1.42** | **0.316** | **17.1** |
| | | ✓ | ✗ | 1.55 | 0.321 | 17.0 |
| | | ✗ | ✓ | 1.83 | 0.356 | 16.1 |
| | | ✗ | ✗ | 1.91 | 0.362 | 16.2 |
| MXFP6e2 | 96 | ✓ | ✓ | **1.29** | **0.305** | **17.4** |
| | | ✓ | ✗ | 1.34 | 0.308 | 17.2 |
| | | ✗ | ✓ | 1.85 | 0.355 | 16.3 |
| | | ✗ | ✗ | 1.91 | 0.358 | 16.3 |
| MXFP4e2 | 128 | ✓ | ✓ | **1.34** | **0.318** | **17.1** |
| | | ✓ | ✗ | 1.41 | 0.326 | 16.9 |
| | | ✗ | ✓ | 1.57 | 0.342 | 16.6 |
| | | ✗ | ✗ | 1.61 | 0.338 | 16.7 |

## 4.5 METHOD EFFICIENCY

This work is designed to leverage emerging hardware Advanced Micro Devices (2025) that supports mixed-precision operations and OCP Microscaling (MX) formats Microsoft et al. (2023). The efficiency improvements of LoRaQ are theoretically well-founded. In this section, we explain the efficiency of our method, which is useful for future hardware implementations that natively support the MXFP data formats.

Previous work used 16-bit activations that must be moved between layers. In LoRaQ, by quantizing the low-rank branch and activations, we have reduced the activations' data movement bandwidth to 8-, 6-, or 4-bit MX data formats. Let's explain the effect of this change on the storage for LoRaQ 's low-rank matrices, $L$ and $R$. In SVDQuant, the low-rank branch includes two weight tensors, $K \times R$ and $R \times N$, while in LoRaQ, it includes two low-rank tensors $K \times R'$ and $R' \times N$. Our study results in Table 4 show that a range of $R'$ could be selected in a co-design scenario, while respecting the bit budget. For example, if MXFP8 is used for the low-rank branch, $R' = 2R$ provides similar weight storage compared to R for an FP16 low-rank branch. The designer can also consider the published specifications for the MX data types Advanced Micro Devices (2025), which indicate 4x FLOPS for MXFP4 vs. FP16 and 2x FLOPS for MXFP8 vs. FP16.

We have also simplified the quantization process by reducing the number of SVD decompositions. In LoRaQ, the low-rank branch calibration requires only one SVD decomposition for initialization. While in SVDQuant, a SVD decomposition is necessary at every calibration iteration. With 100s of calibration iterations, this is a two orders of magnitude reduction in the cost of calculating SVD decomposition.

## 5 CONCLUSION AND FUTURE WORK

This work tackles post-training 4-bit quantization for large diffusion-based transformers, where aggressive precision reduction can severely degrade generative quality. Prior approaches enhance robustness through outlier smoothing or rotations, while recent low-rank error-compensation methods introduce auxiliary full-precision branches. While effective, these methods still require full-precision data movement, incur latency overheads, and necessitate complex calibration, which can be resource-intensive.

We introduce LoRaQ, a novel post-training quantization method that optimizes quantization error by integrating quantized low-rank adaptation branches. This approach eliminates the need for full-precision data movement during inference, while retaining strong error correction capabilities. By design, LoRaQ leverages emerging hardware support for mixed-precision matrix multiplication with sub-16-bit data formats, providing a flexible mixed-precision solution. Our method finds optimized low-rank matrices that are themselves quantized, allowing for fine-grained adjustments to the 4-bit weight matrices. To isolate the benefits of our approach and ensure a fair comparison with prior work, we adopted the same smoothing calibration strategy as SVDQuant. A key advantage of LoRaQ is its model-agnostic nature, which removes the need for a calibration dataset to determine the low-rank matrices. This simplifies the quantization process significantly, in contrast to state-of-the-art methods that often rely on complex and resource-intensive calibration procedures.

Our experimental results demonstrate that LoRaQ consistently improves image quality for 4-bit quantized models across various datasets and architectures. While using the same SINT4 and MX-INT4 data formats as baseline models, our method not only enhances the image quality from 4-bit configurations but also preserves the visual fidelity of the original 16-bit models. Beyond direct comparisons, we investigate the modularity and stability of LoRaQ under various configurations. We explore the impact of quantizing activations in both the low-rank and residual branches to sub-16-bit precision, a critical step for achieving latency improvements on future hardware. Our findings show that with 8-bit activations, LoRaQ exhibits robustness to rank and bit-width trade-offs under fixed memory constraints, highlighting its practical flexibility. Furthermore, we assess the method's limitations with respect to activation precision, demonstrating that it maintains high image quality and similarity with activations quantized down to 6-bit data type for both branches.

Building on these promising results, future work will focus on several key research directions. A primary objective is to develop fused kernels that leverage LoRaQ's heterogeneous compute patterns on specialized hardware such as GPUs and NPUs, which is essential for realizing end-to-end latency improvements. Another priority is to quantize the low-rank activations to further reduce computational overhead, building on our initial findings that full-precision activations are not a strict requirement. We also plan to broaden our evaluations across a broader range of models, datasets, and emerging data formats. Further research will explore adaptive rank and bit-width scheduling, more sophisticated rotation strategies, and calibration-free workflows to continue pushing the boundaries of the quality-efficiency trade-off and accelerate the deployment of quantized models.

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

# A APPENDIX

## A.1 DETAILS: EXPERIMENTS CONFIGURATIONS

### A.1.1 MAIN RESULTS CONFIGURATIONS

**Quantization** Considering Section 4.2, the activations forwarded to the low-rank branch are kept at FP16 (full-precision). The activations forwarded to the residual branch and the residual weights are quantized to a 4-bit data format. SVDQuant keeps the low-rank branch at full-precision for a rank of 32. LoRaQ quantizes the low-rank branch using the same 4-bit data format used for the residual branch, with a rank of 128.

**Smoothing Calibration** Following Li et al. (2025), the smoothing calibration, done prior to any other calibration technique, finds a per-channel vector $\gamma \in \mathbb{R}^b$ where $\forall i \in [0, d[$ :

$$\gamma_i = \frac{\max_j(|\boldsymbol{X}_{j,i}|)^\alpha}{\max_j(|\boldsymbol{W}_{i,j}|)^\beta}$$

where $\alpha$ and $\beta$ are migration strengths (Xiao et al., 2023). The best migration strengths are decided for each layer to minimize the output mean squared error of a predefined module that the quantized layer affects. The SVD with a rank of 32 is used at FP16 precision to quantize the weights of the layer. The evaluation is done on a calibration dataset.

**Optimization** LoRaQ optimizes Equation 4 with Adam (Kingma & Ba, 2014) using a learning rate of $10^{-4}$. The optimization 1000 steps per weight. The rotations in Equation 7 are optimized with a learning rate of $5 \cdot 10^{-1}$ and 500 steps per weight.

## A.2 MIXED PRECISION ANALYSIS AND ABLATION STUDY

**Quantization** For Section 4.3 and Section 4.4, we consider a fully quantized pipeline. In Section 4.3.1 and Section 4.4, the activations forwarded to both branches, low-rank and residual, are quantized to an 8-bit MX data format in. In Section 4.3.2 the activations are quantized to the same data format as the low-rank branch. The residual weights are quantized and fixed to a 4-bit MX data format. Our analysis then considers different configurations for the low-rank matrices.

**Smoothing Calibration** As Appendix A.1.1 we adopt the smoothing calibration prior to LoRaQ. Section 4.3.1 applies LoRaQ with different configuration using the same smoothing scales computes with a low-rank approximation of rank 32. Section 4.3.2 increases the rank for a fixed data format applied to the low-rank matrices. We thus increase the rank used during the smoothing calibration proportionnaly to the rank of the low-rank matrices optimized by LoRaQ.

**Optimization**   We optimize Equation 4 with Adam using a learning rate of $10^{-3}$ for FP formats and $10^{-4}$ for INT format. The optimization 1000 steps per weight. The rotations in Equation 7 are computed with a learning rate of $10^{-1}$ and 500 steps per weight.

### A.2.1   BLOCKWISE FORMATS

In this section, we provide additional details on the block-wise formats used in our experiments. Block-wise formats are essential for efficiently representing and processing the quantized weights in our model. We explore various block sizes and their impact on performance and memory usage.

**SINT4**   is defined by Li et al. (2025) and we use it for comparison between SVDQuant and LoRaQ. The matrices are quantized by block of 64 values with a FP16 scale per block. The rounding applied on scaled values is round-to-nearest (RTN).

**MX Formats**   quantize matrices by block of 32 values with a 8-bit power-of-two scale (8-bit exponent, 0 bit mantissa, or e8m0) (Microsoft et al., 2023). Different formats are used for each scaled and rounded (by RTN) values :

- MXFP8e4 quantizes to a signed e4m3 format;
- MXFP6e2 quantizes to a signed e2m3 format;
- MXFP4e2 quantizes to a signed e2m1 format;

MXINT4 and MXINT8 respectively quantize to a signed int4 and int8 format.

### A.3   PROOFS

### A.3.1   ERROR UPPER BOUND

Using the sub-multiplicativity of the Frobenius norm we show Equation 2:

$$
\begin{aligned}
\mathcal{E}(\boldsymbol{X},\boldsymbol{W}) =& \|\boldsymbol{X}\boldsymbol{W} - Q(\boldsymbol{X})Q(\boldsymbol{W})\|_F \\
=& \|\boldsymbol{X}\boldsymbol{W} - Q(\boldsymbol{X})\boldsymbol{W} + Q(\boldsymbol{X})\boldsymbol{W} - Q(\boldsymbol{X})Q(\boldsymbol{W})\|_F \\
=& \|(\boldsymbol{X} - Q(\boldsymbol{X}))\boldsymbol{W} + Q(\boldsymbol{X})(\boldsymbol{W} - Q(\boldsymbol{W}))\|_F \\
\leq& \|(\boldsymbol{X} - Q(\boldsymbol{X}))\boldsymbol{W}\|_F + \|Q(\boldsymbol{X})(\boldsymbol{W} - Q(\boldsymbol{W}))\|_F \\
\leq& \|\boldsymbol{X} - Q(\boldsymbol{X})\|_F\|\boldsymbol{W}\|_F + \|Q(\boldsymbol{X})\|_F\|\boldsymbol{W} - Q(\boldsymbol{W})\|_F.
\end{aligned}
$$

### A.3.2   DERIVATIVES

In this section, we show that the derivatives exist despite quantization operators to be non differentiable. Since the derivative of the rounding function, $Q$, is zero almost everywhere:

$$
\begin{aligned}
\nabla_{\boldsymbol{LR}}\mathcal{L} &= \nabla_{\boldsymbol{LR}}\|Q(\boldsymbol{W} + \boldsymbol{LR}) - \boldsymbol{W} - \boldsymbol{LR}\|_F^2 \\
&= 2 \cdot \nabla_{\boldsymbol{LR}}(Q(\boldsymbol{W} + \boldsymbol{LR}) - \boldsymbol{W} - \boldsymbol{LR})^T \cdot (Q(\boldsymbol{W} + \boldsymbol{LR}) - \boldsymbol{W} - \boldsymbol{LR}) \\
&= -2(Q(\boldsymbol{W} + \boldsymbol{LR}) - \boldsymbol{W} - \boldsymbol{LR})
\end{aligned}
\tag{8}
$$

Consequently,

$$
\begin{aligned}
\nabla_{\boldsymbol{L}}\mathcal{L} &= \nabla_{\boldsymbol{LR}}\mathcal{L} \cdot (\nabla_{\boldsymbol{L}}\boldsymbol{LR}) \\
&= (-2(Q(\boldsymbol{W} + \boldsymbol{LR}) - \boldsymbol{W} - \boldsymbol{LR})) \cdot (\nabla_{\boldsymbol{L}}\boldsymbol{LR}) \\
&= -2(Q(\boldsymbol{W} + \boldsymbol{LR}) - \boldsymbol{W} - \boldsymbol{LR}) \cdot \boldsymbol{R}^T
\end{aligned}
\tag{9}
$$

Similarly,

$$
\nabla_{\boldsymbol{R}}\mathcal{L} = -2\boldsymbol{L}^T \cdot (Q(\boldsymbol{W} + \boldsymbol{LR}) - \boldsymbol{W} - \boldsymbol{LR})
\tag{10}
$$

### A.4   ADDITIONAL VISUAL RESULTS

Figure 4: Comparison of images generated by PixArt-Σ across different quantization configurations (Section 4.2). Columns show the full-precision model (FP16) against SVDQuant and LoRaQ using both SINT4 and MXINT4 quantization.

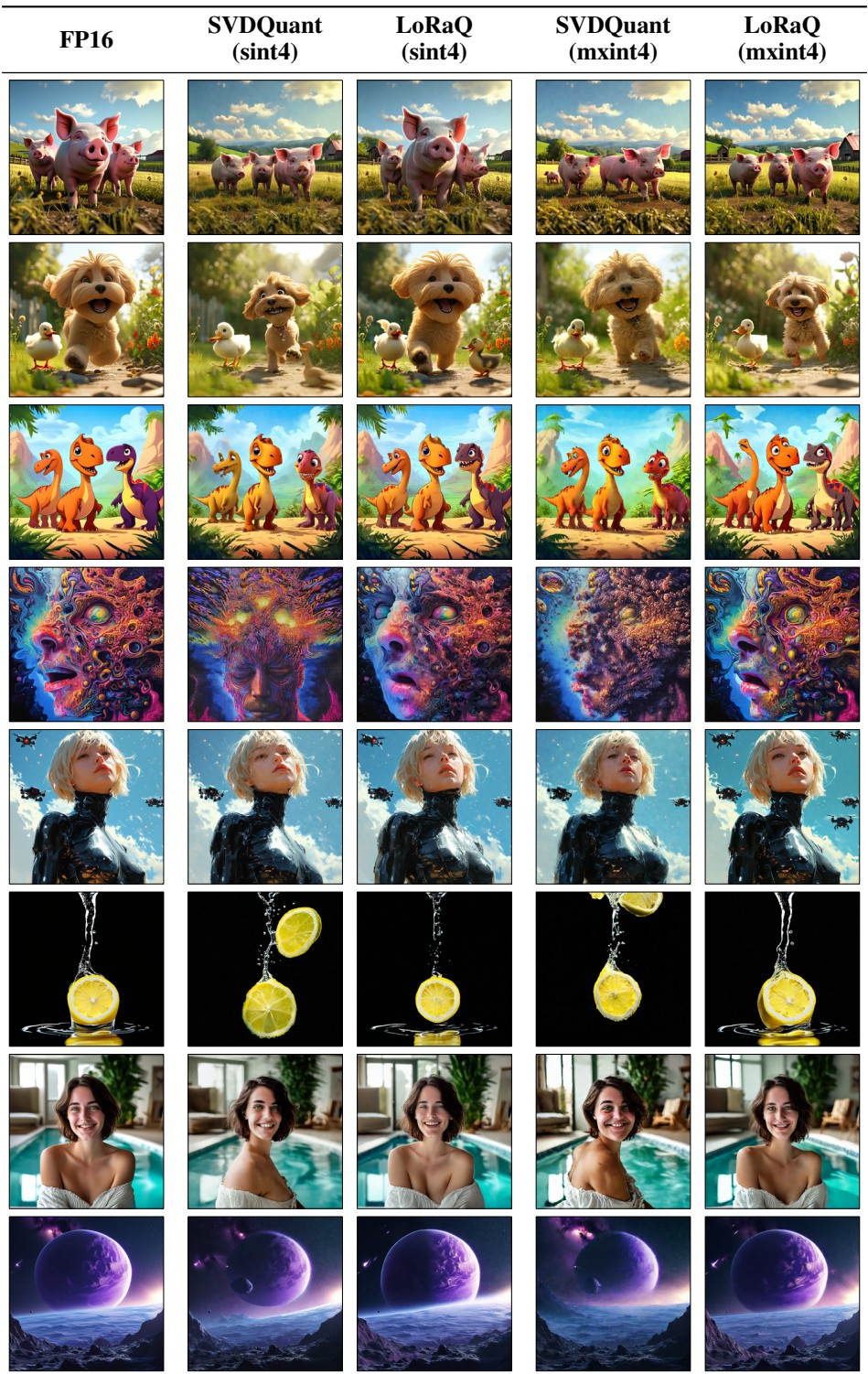

Figure 5: Visual results for different mixed-precision configurations of LoRaQ on PixArt-Σ, corresponding to Table 3. All configurations use MXFP4e2 for residual weights and MXFP8e4 for activations. We vary the rank and data format of the low-rank branch.

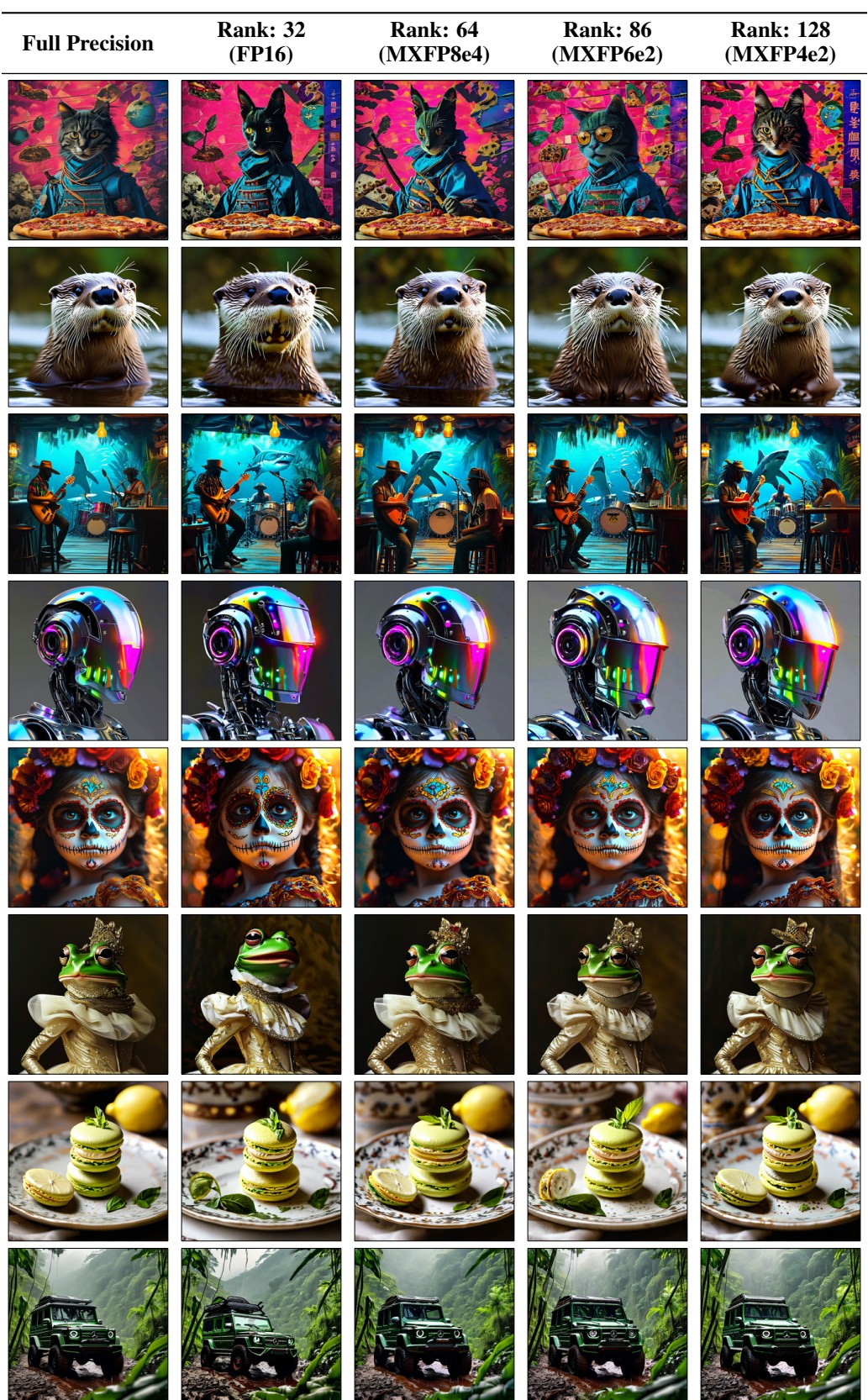

Figure 6: Visual results for different mixed-precision configurations of LoRaQ on PixArt-Σ, corresponding to Table 3. All configurations use MXINT4 for residual weights and MXINT8 for activations. We vary the rank and data format of the low-rank branch.

Figure 7: Visual comparison of LoRaQ on PixArt-Σ, illustrating the impact of rank on generation quality. These results correspond to the quantitative analysis in Table 4. In all configurations, the residual branch is quantized to MXFP4e2, while the low-rank branch and its activations are quantized to MXFP8e4.

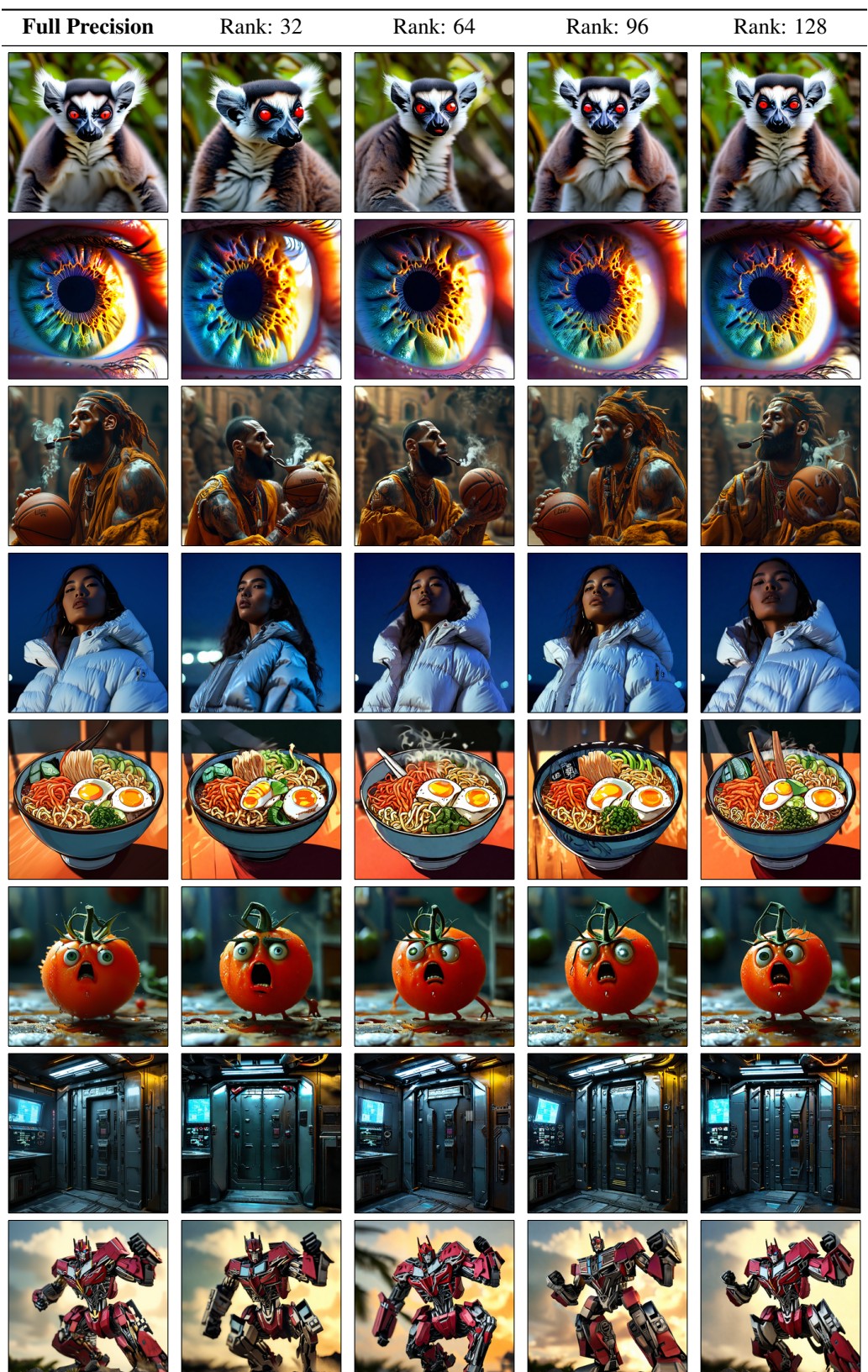

Figure 8: Visual comparison of LoRaQ on PixArt-Σ, illustrating the impact of rank on generation quality. These results correspond to the quantitative analysis in Table 4. In all configurations, the residual branch is quantized to MXFP4e2, while the low-rank branch and its activations are quantized to MXFP6e2.

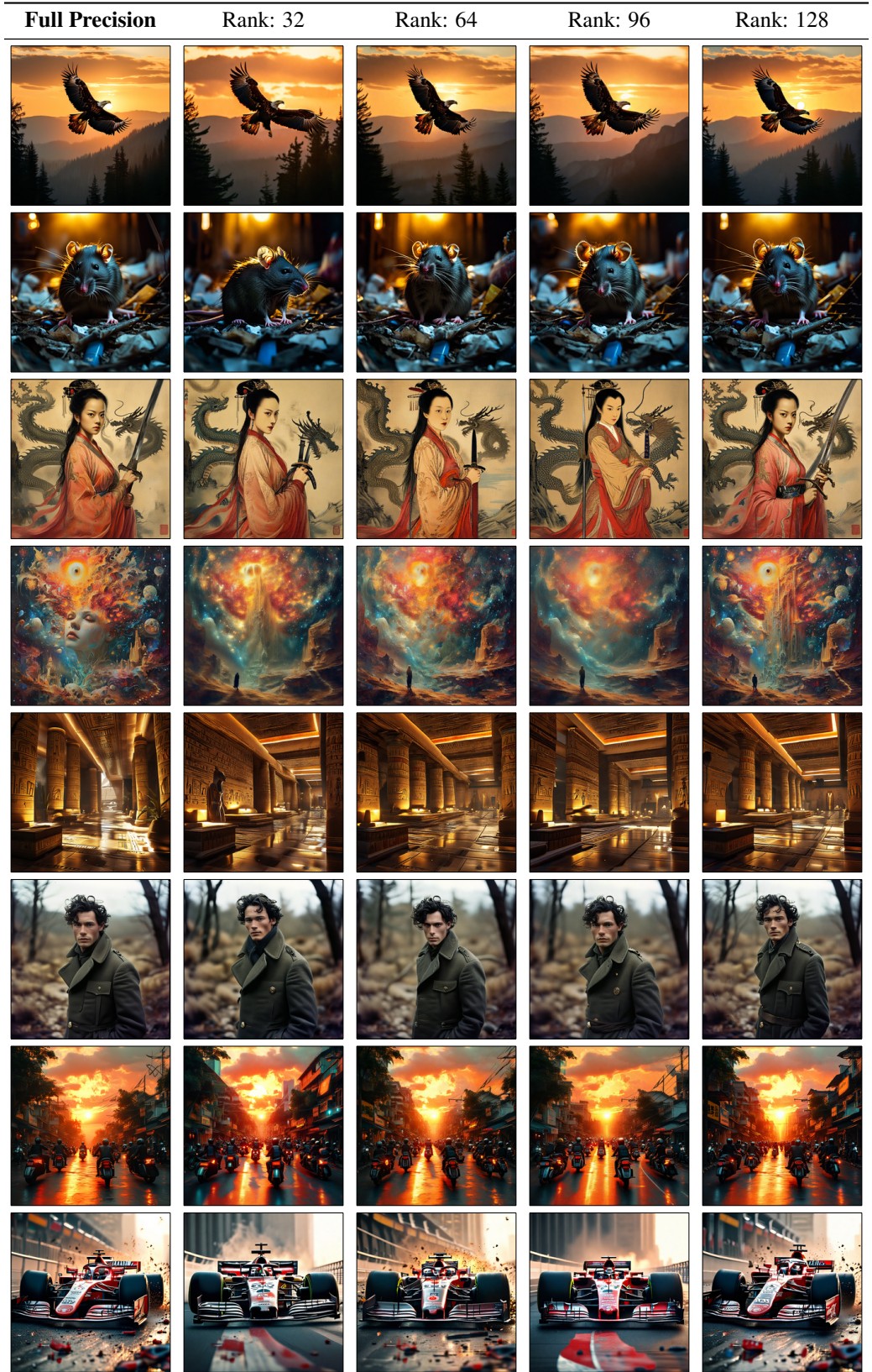

