# OpenReview forum: "LoRaQ: Optimized Low Rank Approximated Quantization Error for 4-bit Quantization"
_ICLR.cc/2026/Conference — Submitted to ICLR 2026_

### Official Review · Reviewer_YVaC · 2025-10-28

**Soundness:** 3
**Presentation:** 3
**Contribution:** 2
**Rating:** 4
**Confidence:** 2

**Summary:**

The paper introduces a framework for low-rank quantization of diffusion models. Instead of performing standard weight quantization (W8A8, W4A8, etc.), the authors decompose each weight matrix into a low-rank factorization (LoRA-style), followed by quantization-aware optimization on the factorized matrices.

**Strengths:**

1. Conceptual Integration of LoRA and Quantization.  The idea of combining LoRA and quantization into a single unified framework is logical and potentially impactful. It effectively exploits the redundancy in diffusion U-Nets and transformer-based blocks.

2. Quantizing diffusion models is increasingly important for deployment on mobile or edge hardware. LoRaQ directly targets this problem, aligning with the trend of compute-efficient generative AI.

**Weaknesses:**

1. The first picture appears to be a non-vectorial image. It is recommended to convert it to a vectorial image.

2. The method is largely an engineering combination of existing paradigms, LoRA-style factorization and quantization-aware training, with a joint optimization loss. While useful, it lacks a novel theoretical component or formal analysis explaining why low-rank factorization improves quantization robustness.

**Questions:**

None

---

> ### Author Response · Authors · 2025-11-26
> **Reply to weaknesses**
>
> Weakness 1:
>
> Thank you for pointing it out. The fully vectorized version of the diagram is available in our revision.
>
> Weakness 2:
>
> We thank the reviewer for the comment and the opportunity to further clarify the role of low-rank factorization in our framework. We borrow the other reviewer’s (hyF4) comment on the strength of our work: although low-rank branches and rotations are known components, their integration into a data-free joint optimization framework that also supports mixed-precision operation is novel. This aligns with our core contribution. LoRaQ jointly optimizes the two low-rank matrices so that the subspace adapts to quantization effects and enables mixed-precision operations within the low-rank branch. We also provide a systematic analysis of the mixed-precision capabilities of a low-rank branch and how emerging micro-scaling formats [1] interact with the branch's rank, a dimension not examined in any prior work to the best of our knowledge, and necessary for designing hardware-compatible hardware strategies [2]. Finally, our intention is not to suggest that low-rank factorization alone inherently improves quantization robustness, but rather to explain how it enables the linear layer to be decomposed into two complementary components: a residual branch with very low-bit weights and a low-rank branch operating on the activation at higher but sub-16-bit precision. This structure allows the low-rank branch to effectively counterbalance the quantization error introduced in the residual path. Established works such as SVDQuant [3] have demonstrated the relevance of low-rank factorization for quantization and the associated theoretical error bounds. In our design, factorization primarily serves as an initialization that inherits these guarantees. We have also added a new section, 4.5, to elaborate on the novelties of our method and its implementation benefits.
>
> References
>
> 1. Open Compute Project Foundation. OCP Microscaling Formats (MX) Specification Version 1.0. Open Compute Project Foundation Technical Specification, September 2023
> 2. Advanced Micro Devices. AMD instinct cdna4 instruction set architecture. Technical report, Advanced Micro Devices, August 2025.
> 3. Muyang Li, et. al., SVDQuant: Absorbing outliers by low-rank component for 4-bit diffusion models. In The Thirteenth International Conference on Learning Representations, 2025

---

### Official Review · Reviewer_hyF4 · 2025-10-29

**Soundness:** 3
**Presentation:** 3
**Contribution:** 3
**Rating:** 6
**Confidence:** 4

**Summary:**

This paper proposes LoRAQ, a data-free calibration approach to minimize the weight quantization error. The work extends from existing low-rank approximation method by proposing mixed-precision configuration with quantized low-rank branch. Quantization errors are further minimized by inserting a learned rotation matrix on the low-rank branch. The proposed method is further accelerated by system implementation optimizations.

**Strengths:**

1. Novelty-wise, this paper moves away from the common data-dependent calibration of block reconstruction and explores a weight-only calibration method. Though the use of low-rank branch and the rotation matrix insertion are well-known ideas, the overall framework remains novel.
2. The exploration on quantizing the low-rank branch opens up a new tradeoff between the rank and the quantization precision of the branch
3. Both quantitative and qualitative results are provided for multiple diffusion models showing the proposed method outperforming SVDquant baseline.
4. The paper has a clear presentation overall, easy to follow.

**Weaknesses:**

1. For the efficiency evaluation, though the paper claims improved hardware support by removing floating scales and micro-scaling formats, no real runtime measurements are provided in the evaluation to demonstrate the improved efficiency. A latency or throughput comparison needs to be conducted to justify the improvement.
2. The paper proposes multiple techniques, such as the new data-free calibration strategy, adding rotation to the low-rank branch, and performing different format of quantizations on the low-rank branch. However, ablation study is lacking to show the effect of each individual treatment. Ablation is especially needed to show the performance gain brought by the different calibration strategy and the rotation matrix.
3. The paper claims a fiar comparison with SVDquant. However, the proposed method utilizes additional rotation matrix, which may add additional overhead to the inference.

**Questions:**

Please provide additional results to tackle the three weaknesses mentioned in the previous section.

---

> ### Author Response · Authors · 2025-11-26
> **Answer to weaknesses**
>
> Weakness 1:
>
> We appreciate the reviewer’s observation regarding the absence of real runtime measurements to substantiate the claimed efficiency gains.
> This work is designed to leverage emerging hardware (such as AMD MI355 [1]) that supports mixed-precision operations and OCP Microscaling (MX) formats [2].
> Hence, the runtime evaluation was not feasible due to a lack of access to supportive hardware.
> However, the efficiency improvements of LoRaQ are theoretically well-founded, and we have added details in section 4.5 to explain them. We have described the advantage of quantizing activations in LoRaQ to 8-, 6-, or 4-bit MX data formats vs.
> SVDQuant’s 16-bit activations, thereby reducing data movement bandwidth. We have elaborated on the differences in weight storage between SVDQuant and LoRaQ and explained the design flexibility provided in LoRaQ. A simple comparison could be done by looking at a linear layer with $M\times K$ activations and $K\times N$ weights. In SVDQuant, $M\times K$ activation tensor has two bytes per element (FP16), while in LoRaQ, the same activation tensor has its elements encoded with MX data types (FP8/6/4).
> SVDQuant decomposes the weight tensor into a $K\times N$ 4-bit residual and two low-rank weight tensors, $K\times R$ and $R\times N$.
> LoRaQ will decompose the same weight tensor into a $K\times N$ MXFP4 tensor and two
> low-rank tensors $K\times R'$ and $R'\times N$.
> Our study results in Table 4 show that a range of $R'$ could be selected in a co-design scenario, while respecting the bit budget.
> For example, if MXFP8 is used for the low-rank branch, $R' = 2R$ provides similar weight storage compared to R for an FP16 low-rank branch. We have provided different rank values per data format in Table 4. The designer can also consider the published specifications for the MX data types [1], which indicate 4x FLOPS for MXFP4 vs. FP16 and 2x FLOPS for MXFP8 vs. FP16.
>
> References
> 1. Advanced Micro Devices. AMD instinct cdna4 instruction set architecture. Technical report, Advanced Micro Devices, August 2025.
> 2. Open Compute Project Foundation. OCP Microscaling Formats (MX) Specification Version 1.0. Open Compute Project Foundation Technical Specification, September 2023
>
> Weakness 2:
>
> We find this point helpful in strengthening our paper. Therefore, we have conducted more experiments and have provided them in Table 5. We have added six more rows to show the effect of calibration with and without the rotation matrices. And have explained our new results in section 4.4.
>
> Weakness 3:
>
> In LoRaQ, the rotation matrices are applied offline during optimization and fully fused into the low-rank factors, so only the rotated low-rank matrices remain at inference. This introduces a small offline optimization cost that improves quantization quality (Table 5). However, already-rotated weights are used at inference, which adds no memory, runtime, or parameter overhead, resulting in no extra complexity during inference. Therefore, the use of rotations does not affect the fairness of the comparison. We thank the reviewer for raising this point. We have updated the text for Figure 1 to make this point clearer.

---

> ### Comment · Reviewer_hyF4 · 2025-11-27
>
> I would like to thank the author for the replies and updates. I take your explaination on your efficiency claim.
>
> Meanwhile, I still have a question on the ablation study. You claim in the abstract that "data based calibration contributes to the complexity of the quantization process and involves risks such as potential accuracy degradation", However, there seems to be no results directly supporting this claim. Is it possible to add the activation $X$ into the calibration objective of Equ. (7)? Or are these existing evidence showing the weight-only calibration to be more effective than the data-dependant calibration process?

---

> > ### Author Response · Authors · 2025-12-03
> > **Abstract Correction**
> >
> > We thank the reviewer for raising this point. The sentence in the abstract unintentionally suggested that data-based calibration introduces accuracy degradation while data-free calibration does not. This implication is inaccurate and not supported by our experiments. We apologize for the imprecise wording.
> >
> > We have updated the abstract to remove this implication and now focus solely on the computational complexity of data-based calibration. This correction does not affect the contributions or conclusions of our work.
> >
> > > In addition, data-based calibration contributes to the computational complexity of the quantization process, especially because search policies must evaluate many parameter configurations using a small calibration subset.
> >
> > Additionally, Equation (7) is specifically designed for our data-free calibration method, and incorporating activations would convert it into a data-dependent procedure.

---

### Official Review · Reviewer_RNYk · 2025-11-01

**Soundness:** 4
**Presentation:** 4
**Contribution:** 4
**Rating:** 10
**Confidence:** 1

**Summary:**

I am unable to assess this paper and have alerted the ACs to seek an opinion from different reviewers on 14 Oct 2025.

**Strengths:**

I am unable to assess this paper and have alerted the ACs to seek an opinion from different reviewers on 14 Oct 2025.

**Weaknesses:**

I am unable to assess this paper and have alerted the ACs to seek an opinion from different reviewers on 14 Oct 2025.

**Questions:**

I am unable to assess this paper and have alerted the ACs to seek an opinion from different reviewers on 14 Oct 2025.

---

### Official Review · Reviewer_7mZU · 2025-11-18

**Soundness:** 2
**Presentation:** 3
**Contribution:** 3
**Rating:** 6
**Confidence:** 2

**Summary:**

This paper proposes LoRaQ, a data-free calibration approach to optimize quantization error compensation. With W4A4 settings, LoRaQ achieves optimal experimental performance compared to existing methods.

**Strengths:**

1. The results show that the proposed method significantly improves the metrics compared to existing SVDQuant methods, demonstrating its advantages.
2. The paper claims that they will release the PTQlibrary for transformer blocks, which will contribute to future research.

**Weaknesses:**

1. The paper mentions that a major advantage of LoRaQ is its model independence, which eliminates the need to calibrate datasets to determine low-rank matrices. This significantly simplifies the quantization process. However, the authors do not explain the performance gains resulting from this simplification. Are there any metrics that can quantify the benefits brought by the model?
2. The paper only conducted experiments on the PixArt-Σ and SANA models. Is it generalizable for other models with different architectures or different numbers of parameters?

**Questions:**

Please see Weaknesses

---

> ### Author Response · Authors · 2025-11-26
> **Answer to weaknesses**
>
> Reply to weakness 1:
>
> Thank you for your critical question. We have added section 4.5 to address this question. Regarding simplifications to the quantization process, the data-free low-rank calibration we designed (Equation 7) is resource-efficient and optimizes one layer at a time independently with batch-parallelism, providing significant flexibility in resource usage. In LoRaQ, the low-rank branch calibration requires only one SVD decomposition for initialization.  While at SVDQuant, a SVD decomposition is necessary at every calibration iteration. Considering the number of iterations in SVDQuant in the 100s, this is two orders of magnitude reduction in the cost of calculating SVD decomposition.
>
> Reply to weakness 2:
>
> We appreciate the reviewer for this question. As discussed in the paper, improving generalization is a key direction for future work. Our method operates directly on linear layers, including both explicit linear layers and $1\times1$ convolutions (as in the SANA architecture). Our calibration method reduces the loss value, local to the layer, and is tested in models with 600M and 1.6B parameters. Due to our compute and time budgets, we did not explore larger models, which can be added to the experiment list at a later time. However, given the plug-and-play nature of our method and its generality, we expect our approach to transfer naturally to different and larger models.

---

> > ### Comment · Reviewer_7mZU · 2025-11-26
> >
> > Thanks for your reply, which partially solved my concern. I will maintain my positive score.

---

### Meta-Review · Area_Chair_5ELA · 2025-12-24

**Summary:**

The paper proposes a method for 4-bit weight quantization of diffusion transformers. Comparing with SVDQuant, it solely focuses on weight quantization, and the main idea is: 1) use low-rank to absorb the quantization error rather than original weight; 2) quantize low rank component for memory saving/speedup; 3) add rotation to the low rank component to reduce Q2 quantization error. The method is evaluated on PixArt and SANA, showing superior quality than SVDQuant.

Concerns include lack of novelty, extension to other models, and lack of wall-clock latency results. AC thinks the concerns are either not fully addressed, or are indeed inevitable. Therefore, though the paper has some merits, the quality is still slightly below the bar.

**Reviewer Concerns:**

1. lack of wall-clock latency results (hyF4) / fair compute comparison with svdquant (hyF4)
While reviewer buys in the response. AC is curious why the authors claim that MXFP hardware is not available, as Blackwell hardware is already quite abundant. If I am understanding correctly, LoraQ uses a higher rank of 128 than SVDQuant (32). Therefore, it is not surprising that LoraQ is better. At least there should be some timing result showing that the 128*MXFP8 would not be slower than 32*FP16/BF16.

2. lack of novelty (yvac)
While the author argued on the novelty, the consideration remains. Equation (3-4) can be considered novel. Other aspects, including quantizing weight by minimizing reconstruction error, quantized low-rank branch, and rotation, should be viewed as incremental.

3. extension to other models (ymZU)
This concern remains.

**Reviewer Scores:**

Reviewers originally has a rating of (6, 6, 4), note that there is an invalid review with a rating of 10. I think reviewer will retain their ratings.

---

### Decision · Program_Chairs · 2026-01-26

Reject